# Regulatory properties of transcription factors with diverse mechanistic function

**Md Zulfikar Ali**[1,2,3], **Sunil Guharajan**[1,2], **Vinuselvi Parisutham**[1,2], **Robert C. Brewster**[1,2]*

**1** Department of Systems Biology, University of Massachusetts Medical School, Worcester, Massachusetts, United States of America, **2** Department of Microbiology and Physiological Systems, University of Massachusetts Medical School, Worcester, Massachusetts, United States of America, **3** Department of Geology, Physics and Environmental Science, University of Southern Indiana, Evansville, Indiana, United States of America

* Robert.Brewster@umassmed.edu

## Abstract

Transcription factors (TFs) regulate the process of transcription through the modulation of different kinetic steps. Although models can often describe the observed transcriptional output of a measured gene, predicting a TFs role on a given promoter requires an understanding of how the TF alters each step of the transcription process. In this work, we use a simple model of transcription to assess the role of promoter identity, and the degree to which TFs alter binding of RNAP (stabilization) and initiation of transcription (acceleration) on three primary characteristics: the range of steady-state regulation, cell-to-cell variability in expression, and the dynamic response time of a regulated gene. We find that steady state regulation and the response time of a gene behave uniquely for TFs that regulate incoherently, i.e that speed up one step but slow the other. We also find that incoherent TFs have dynamic implications, with one type of incoherent mode configuring the promoter to respond more slowly at intermediate TF concentrations. We also demonstrate that the noise of gene expression for these TFs is sensitive to promoter strength, with a distinct non-monotonic profile that is apparent under stronger promoters. Taken together, our work uncovers the coupling between promoters and TF regulatory modes with implications for understanding natural promoters and engineering synthetic gene circuits with desired expression properties.

## Author summary

In this study, we explored the regulatory roles of TFs in gene expression by developing a model that considers the effects of two parameters: stabilization (interactions that modify binding of RNA polymerase to the promoter) and acceleration (interactions that modify the rate of transcription initiation). By applying this model, we evaluated how these factors influence gene expression dynamics, noise, and response times. We show that different combinations of stabilization and acceleration can lead to a rich diversity of regulatory properties. The findings offer valuable insights into the complex interactions between TFs and promoters, underscores the need to characterize TFs in terms of their regulatory

**Data Availability Statement:** Code and results are available on Github: https://github.com/zulfikgp/TFFunction.

**Funding:** RCB received funding from 1R35GM128797, National Institute of General

Medical Sciences, https://www.nigms.nih.gov/. The funders had no role in study design, data collection and analysis, decision to publish, or preparation of the manuscript.

**Competing interests:** The authors have declared that no competing interests exist.

modes, and potential guides the engineering of synthetic circuits with desired expression characteristics.

## Introduction

Gene regulation is an essential process that controls a cell's response to both external and internal stimuli. Accordingly, the ability to predict gene expression levels using forward modeling approaches has been a focus of in the field of regulatory biology [1–5]. These models have the potential to impact a wide range of applications, from biomedical—where deciphering the regulatory genome may clarify the influence of mutations—to synthetic biology, where constructing circuits with robust, predictable responses facilitates biological designs with industrial and therapeutic applications. However, achieving a model that is broadly useful has been a notoriously difficult problem, despite the breadth of data accumulated from both natural and synthetic genes [4, 6–13]. Recently, the need for precise quantitative models of gene input-output functions has been highlighted by studies demonstrating the sensitivity of essential regulatory networks to transcription factor (TF) dosage [14–17].

The regulation of a specific promoter hinges on a complex interplay of factors including the number and position of TF binding sites, the RNA polymerase (RNAP) recognition sequence, and the concentration and activity of TFs in a given cellular context. The actual regulatory function of a transcription factor (TF) once it binds to the promoter is a crucial detail that is challenging to infer, with TFs classified as "activators" or "repressors" based on their overall regulatory function on a specific promoter. However, it has become evident that these labels alone are insufficient for predicting the role of a TF on a different gene or in a different context; there are countless examples of TFs that act at similar binding sites on different promoters with distinct regulatory function [18, 19]. One reason for this is that the mechanics of regulation can affect various kinetic steps of the transcription cycle, as evidenced by studies [20–25].

The role of specific TFs on distinct steps of RNAP-promoter dynamics has paved the way for further questions on the implications of such kinetic control [26–33]. Certain TFs have been found to regulate transcription by affecting the ability of RNAP to occupy the promoter, achievable either through steric hindrance [2, 34] or by forming energetically favorable interactions with RNAP [35–37]. On the other hand, other TFs act on the kinetic steps of open complex formation and promoter escape [25, 27]. Furthermore, it has been shown that individual TFs may impact both rates [38, 39], with recent studies having formalized a model for gene regulation that accommodates regulation on these two steps of the transcription process (Fig 1A) [18, 22, 39, 40].

In this work, we use a model that generalizes the regulation process across both steps, makes no assumptions about the net role of any TF, and takes into account that a TF can influence one or both kinetic steps of transcription. TF function is quantified through two parameters that measure the fold-change in the likelihood of RNAP occupancy at the promoter or the rate of initiation when TF is bound as compared to these processes when TF is not bound. The model refers to changes in RNAP occupancy at the promoter as stabilizing/destabilizing (parameterized as $\beta$—stabilizing if $\beta > 1$, destabilizing if $\beta < 1$), and changes to the rate of open complex formation as accelerating/decelerating (parameterized by $\alpha$—accelerating if $\alpha > 1$, decelerating if $\alpha < 1$). In previous research, we utilized this model to measure the regulatory parameters of the *E. coli* TF, CpxR, in relation to binding position and found $\alpha$ and $\beta$ values, and thus the net regulatory role of the TF, are strongly dependent on the binding location.

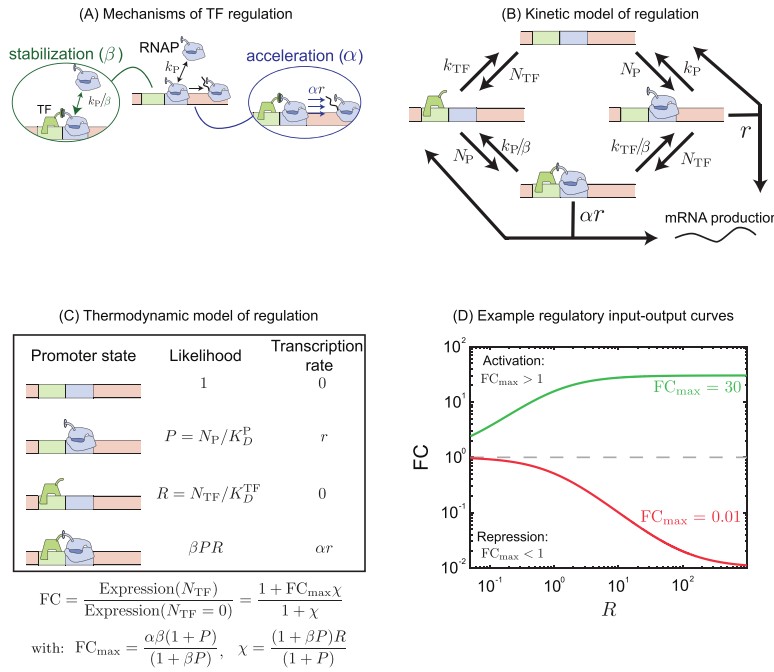

**Fig 1. Schematic of regulation by TF in thermodynamic and kinetic model.** (A) In this model the TF may function on two distinct steps of transcription related to the stabilization or destabilization of polymerase at the promoter and the acceleration or deceleration of transcription initiation by bound polymerase. (B) A kinetic model describing the transitions of the gene between 4 promoter states: empty, polymerase bound, TF bound and both TF and polymerase bound. Regulation is described by modifications to the rates of transition described here as $\alpha$ and $\beta$. (C) A simplified equilibrium model of the same system using the same parameters. (D) Example regulation curves for a TF which has an activating interaction $\text{FC}_{\text{max}} > 1$ (green curve) and for a TF which has a repressive interaction, $\text{FC}_{\text{max}} < 1$ (red curve).

We consider the implications of this model using the simplest regulatory architecture, one where the promoter is regulated only by a single TF and that TF has only one binding site on the promoter. We use this model of gene regulation to explore the implications for regulation by a single TF in terms of three distinct qualities: the spectrum of responses a TF can have to different promoters at the mean expression level, the noise in expression, and the dynamic response time of a regulated gene. Using thermodynamic and kinetic model approach we find that each of these properties depends sensitively on the proportion of each regulatory mechanism used by the regulating TF. Below we discuss the rich possibilities of gene regulatory responses and dynamics that result from the simple assumption that TFs are capable of regulating both the RNAP occupancy step and the transcription initiation steps of gene expression.

## Results

### Model

The full kinetic model of gene regulation by a single TF is shown in Fig 1B. The model accounts for binding of TF and polymerase independently at rates that are proportional to the free TF ($N_{\text{TF}}$) or polymerase concentration ($N_{\text{P}}$). The TF and the polymerase unbind/dissociate from the bound states at the rate that is independent of the TF/polymerase concentration and is only dependent on the interaction between the TF and the polymerase, and binding site identity. The polymerase may also unbind through a productive initiation event where it will

create an mRNA/protein. The regulatory role of the TF is encoded in two ways. The first, which we call stabilization, is represented as a constant factor, $\beta$, that alters the rate of TF and polymerase dissociation when co-bound. The second regulatory mechanism, acceleration $\alpha$, is a constant multiplicative factor that modulates the initiation rate, $r$. These factors fall in the range between 0 and $\infty$; values greater than 1 represent regulatory interactions that promote gene expression (faster initiation or greater polymerase occupancy at the promoter), while values less than 1 represent regulatory interactions that repress expression (slower initiation or lower polymerase occupancy at the promoter). For simplicity and keeping the model tractable, we assume that the mRNA and protein production rates are incorporated in the initiation rate $r$ and we do not include mRNA dynamics explicitly in the model.

First, we discuss the thermodynamic framework of the above-mentioned model where we compute the equilibrium probability of all the four possible states of gene expression namely, free, polymerase bound, TF bound, and co-bound states as shown in Fig 1C [35, 41–46]. The advantage of this approach is that it tends to produce tractable analytic results at the cost of not explicitly considering if occupancy occurs through modulation of binding or unbinding rates of TFs or polymerases. The net fold-change, gene expression in the presence of TF divided by expression in the absence of TF, predicted from this model is a function of the regulatory parameters, $\alpha$ and $\beta$, the number of TFs and polymerase present, $N_{\mathrm{TF}}$ and $N_{\mathrm{P}}$ as well as the affinity of the TF and polymerase for their binding sites, $k_D^{\mathrm{TF}}$ and $k_D^{\mathrm{P}}$. In this model we can write the fold-change, FC [39],

$$\mathrm{FC} = \frac{1 + \chi \mathrm{FC}_{\mathrm{max}}}{1 + \chi}, \tag{1}$$

$$\text{with } \mathrm{FC}_{\mathrm{max}} = \frac{\alpha\beta(1 + P)}{(1 + \beta P)} \text{ and } \chi = \frac{(1 + \beta P)}{(1 + P)} R, \tag{2}$$

where $P = N_{\mathrm{P}}/k_D^{\mathrm{P}}$ and $R = N_{\mathrm{TF}}/k_D^{\mathrm{TF}}$. The fold-change curve can be neatly arranged into two parameters. First, $\chi$ is the effective concentration of the TF and depends on how many TFs are in the system compared to the equilibrium constant. Additionally, there is a contribution from stabilization regulatory interactions. Positive interactions between TF and polymerase ($\beta > 1$) provide an apparent increase in effective TF concentration by forming stable complexes at the promoter which increase the occupancy of TF at its binding site compared to occupancy in the absence of those interactions. Similarly, destabilizing interactions ($\beta < 1$) can reduce the occupancy by hindering or preventing co-occupancy. Since this effect also requires polymerase bound, the magnitude also depends on $P$. The other parameter, $\mathrm{FC}_{\mathrm{max}}$ is the fold-change of the gene when TFs are saturating; *i.e.* when $\chi \gg 1$. In previous work [39], we wrote a similar relationship but under the assumption of a weak promoter ($P \ll 1$). Here we do not make this assumption.

The model is greatly simplified under the assumption that the promoter is weak both without TF ($P \ll 1$) and with TF (($\beta P \ll 1$); in this case, the parameters simplify greatly to $\chi = N_{\mathrm{TF}}/k_D^{\mathrm{TF}}$ and $\mathrm{FC}_{\mathrm{max}} = \alpha\beta$. This result is intuitive, in this limit the maximum amount of regulation $\mathrm{FC}_{\mathrm{max}}$ is the two mechanisms multiplied (how much more likely polymerase is to be at the promoter times how much faster it initiates transcription) and the effective number of TFs in the system $\chi$ is just the number of TFs divided by the equilibrium dissociation constant for the TF. In this limit, the two regulatory parameters and the two modes are not separable in their contribution to the average expression level of the gene. Example curves are shown in Fig 1D that show stereotyped responses for repression (red) and activation (green) regulatory

interactions. In all cases, the fold-change is monotonically increasing (activation) or decreasing (repression) until it approaches $FC_{max}$.

## A phase space approach to visualizing regulatory function

One useful way to portray TF function is by plotting the regulatory function of a TF as a function of $\alpha$ and $\beta$. This is shown in Fig 2A with contour lines that mark $FC_{max}$ in increments of 100–fold. The black contour represents $FC_{max} = 1$ which bifurcates the graph between activators (above the black line) and repressors (below the black line). The phase space is shown for two values of $P$, a weak promoter (left, $P = 0.01$) and a strong promoter (right, $P = 1$). When $P$ is small, these lines simply follow the relationship $\beta = FC_{max}/\alpha$ as indicated above which creates a straight line in log-log space corresponding to the weak limit described earlier. The lines only deviate from this relationship when $\beta P$ is no longer significantly less than 1 or equivalently when $\beta \gg 1$. However, when $P$ increases we see divergence from this simple relationship in the upper half of the plot where increasing $\beta$ no longer increases the magnitude of the regulation (*i.e.* the contours become vertical).

Fig 2B shows contours of $FC_{max}/\alpha$ for different values of $P$. As $\beta$ becomes larger, $FC_{max}$ simply becomes $\alpha(1 + P)/P$ and $FC_{max}$ saturates as the occupancy of polymerase at the promoter goes to one (Fig 2C). Furthermore, it can be seen that when $\beta > 1$ increasing $P$ leads to a monotonic decrease in $FC_{max}$, however when $\beta < 1$ we see the opposite, $FC_{max}$ monotonically

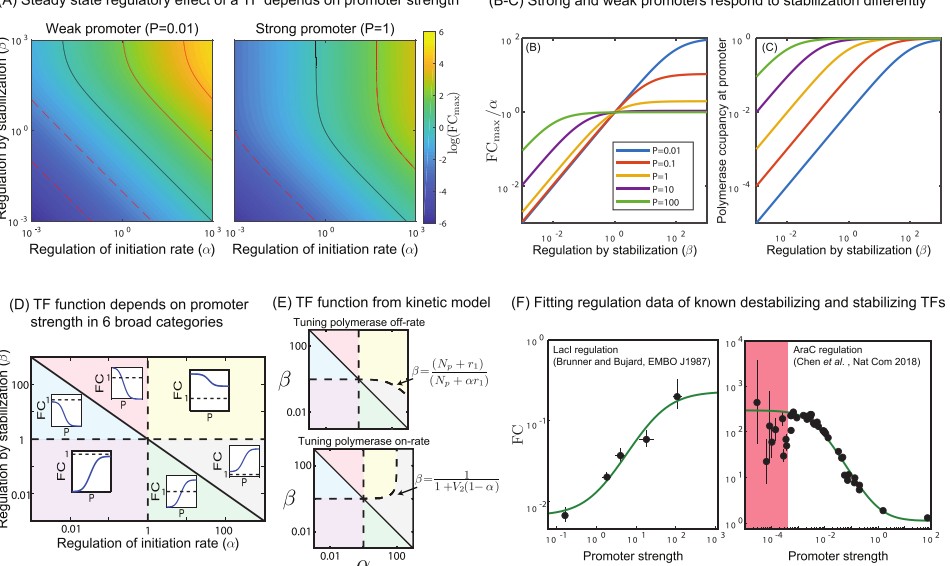

**Fig 2. TF function predicted from regulatory parameters $\alpha$ and $\beta$.** (A) The net "maximum" regulation from a TF as a function of the parameter $\alpha$ and $\beta$ for a weak promoter ($P = 0.01$, left panel) and a strong promoter ($P = 1$, right panel). Black contours mark $FC_{max} = 1$ and subsequent contours represent 100–fold increase or decrease in $FC_{max}$. The observed regulation from a TF depends on promoter strength. (B-C) Positive stabilization is ineffective on strong promoters due to saturation of polymerase at the promoter. (D) The regulatory parameter space ($\alpha$-$\beta$ space) splits into 6 regions based on how a gene will respond to titration of promoter strength. The three lines that divide this space correspond to $\alpha\beta = 1$, $\alpha = 1$ and $\beta = 1$. Insets in each region show the qualitative behavior of TFs from that region of parameter space as $P$ is titrated. The black dashed line in each inset indicates when fold-change equals 1 (E) The space is altered slightly if solved in the kinetic model; the boundary corresponding to $\beta = 1$ is now more complex depending on if the on-rate or off-rate of polymerase is tuned to change promoter strength. Here $V_2$ is the ratio of the transcription initiation rate and the polymerase unbinding rate. (F) Fit of LacI repression (left) and AraC activation (right) data from [47] and [48] using the general model outlined here. For these TFs, we know the primary mode of regulation is through strong destabilization [49] and strong stabilization [50], respectively.

increases. This feature combined with the fact that phase-space of activation and repression varies with $P$ generates an intricate pattern in phase-space.

**Dependence of regulatory outcome on promoter strength.**   The interdependence of fold-change on regulatory parameters of TF ($\alpha$ and $\beta$) and promoter strength ($P$) divide the $\alpha - \beta$ space into a six regions by the lines $\alpha = 1$, $\beta = 1$ and $\alpha\beta = 1$. This is diagrammed in Fig 2D. Importantly, each region has a particular regulatory behavior when $P$ is tuned independent of the TF concentration. The six regions respond to increasing values of $P$ as follows

1. $\alpha\beta < 1$ and $\beta > 1$: low repression to high repression (blue region)

2. $\alpha\beta > 1$ and $\alpha < 1$: activation to repression (red region)

3. $\alpha\beta < 1$ and $\alpha > 1$: repression to activation (green region)

4. $\alpha\beta > 1$ and $\beta < 1$: low activation to high activation (gray region)

5. $\alpha < 1$ and $\beta < 1$: high repression to low repression (purple region)

6. $\alpha > 1$ and $\beta > 1$: high activation to low activation (yellow region)

It is important to note that this behavior is independent of the TF concentrations and solely depends on the TF regulatory parameters and the promoter strength. The most interesting behavior we observe is the switching of regulation from activation to repression in the red region, and repression to activation in the green region as $P$ is tuned. The promoter strength at which the switch occurs is predicted to be $P_s = (\alpha\beta - 1)/(\beta - \alpha\beta)$. As a result, in the red shaded region, when $P < P_s$ the TF behaves as an activator and when $P > P_s$ as a repressor and vice versa for the green region. There is a simple qualitative way to understand this phenomenon. In the red region the regulatory interactions help stabilize polymerase at the promoter but also slow the transcription initiation process when polymerase is bound. For a weak promoter, this trade-off can result in a net increase in expression if $\beta$ is greater than $1/\alpha$ because the reduction in initiation is compensated by recruiting more polymerase at the promoter. For instance, as an example if $\beta = 10$ and $\alpha = 1/2$, polymerase will occupy a weak promoter 10 times more which even if the TFs presence slows transcription by 1/2, this is a net activation. On the other hand, for strong promoters which may gain little or no additional polymerase occupancy from stabilizing interactions, the net change is that all transcription is slowed by 1/2 resulting in repression. It can also be seen from Eqn. 2 that when $P \rightarrow 0$ the $\text{FC}_{\text{max}}$ becomes $\alpha\beta$. This means any reduction in fold-change due to reduction in initiation ($\alpha < 1$) is compensated by an increase in the polymerase occupancy by stabilization. On the contrary when $P \rightarrow \infty$ the $\text{FC}_{\text{max}}$ goes to $\alpha$ which is independent of the stabilization factor $\beta$. Consequently, $\alpha < 1$ always leads to repression irrespective of the contribution from the stabilization. In both cases, we expect this phenomenon for TFs that are "incoherent" in their regulation; *i.e.* they slow one step while speeding up another but the product of these effects $\alpha\beta > 1$.

**Implications of the full kinetic model.**   The thermodynamic model outlined in Fig 1C is agnostic to how stabilization $\beta$ is incorporated in the model and also whether $P$ is tuned through polymerase number affecting the net binding rate or the unbinding rate. To explicitly include these, and non-equilibrium effects, we turn to the full kinetic model outlined in Fig 1B and solve for features such as the mean level of expression, the response time of single cells and the noise in gene expression using a chemical master equation approach (see S1 Text section I). The expression for $\text{FC}_{\text{max}}$ in the full model is more complex, given by

$$\text{FC}_{\text{max}} = \frac{\alpha\beta(1 + P + V_2)}{1 + P\beta + \alpha\beta V_2}, \tag{3}$$

with a cumbersome expression of fold-change for finite TF concentrations ($S1$ Text section I). Here, $V_2 = r/k_{\text{off,P}}$ is the ratio of the transcription rate from polymerase bound state and the polymerase unbinding rate without TF ($k_{\text{off,P}}$). In the limit $V_2 \rightarrow 0$ or alternatively when the transcription rate is much slower than the polymerase unbinding rate, the expression for $FC_{\text{max}}$ reduces exactly to the thermodynamic model expression. This is intuitive because in the thermodynamic model, there is no transition of states due to initiation of transcription and the total gene expression is found by summing the product of equilibrium probabilities of different states with their corresponding transcription rate. As such in the full kinetic model, when the transcription rate is very slow, the transition between states become rare events. The phase space of Fig 2D changes in the full kinetic model. The phase-space is divided in the thermodynamic model by the lines $\alpha = 1$, $\beta = 1$ and $\alpha\beta = 1$. In the full kinetic model, the boundaries of $\alpha = 1$ and $\alpha\beta = 1$ remain unchanged, however, the third line changes depending on how $P$ is tuned, *i.e.* by changing promoter affinity for polymerase $k_{\text{off,P}}$ or by changing concentration of polymerase $k_{\text{on,P}}$. Fig 2E shows both cases of these spaces. As can be seen, when $\alpha \leq 1$ the lines between regions of space are unchanged (the red, blue and purple shaded regions). However, when $P$ is tuned by changing the off-rate of polymerase from the promoter, the yellow region (activation becoming weaker) grows at the expense of the gray region (activation becoming stronger). However, the opposite is true when $P$ is tuned by changing the on-rate of polymerase where the yellow region now shrinks at the expense of the gray region. The analytic expression for this line can be solved exactly and is shown next to each plot (also see Table 1). All results that follow will be derived from the full kinetic model.

**Applications of general model to data.** In Fig 2F, we plot measured fold-change against strength of promoter for two different TFs. The first, plotted on the left, is data for LacI binding to the O1 operator at the natural proximal binding site (centered at + 11) to regulate several distinct promoters. In this case, *in vitro* measurements have shown that LacI binding occludes RNAP binding almost completely [49]. This corresponds with $\beta \ll 1$ in our model. The left panel of Fig 2F shows *in vivo* measurements of fold-change (from [47]) against the strength of each promoter (from *in vitro* $K_{\text{eq}}$ measurements [51]). We find that for the best fit, $\beta = 0.029$, consistent with the reduction of specific RNAP binding to the promoter being reduced to background levels in reference [49]. Further, we find that $\alpha = 0.23$, which is to say that we infer that even in the rare case where RNAP can co-bind, initiation is reduced substantially. The right panel in Fig 2F shows measurements of AraC regulation on a library of promoter with a wide range of constitutive expression levels from Chen *et al* [48]. AraC is known to stabilize RNAP at the promoter and promote open complex formation, which in our model corresponds to $\beta > 1$. In this case, we find 194-fold stabilization fits the data with only a small change to the initiation rate, $\alpha = 1.20$. The red shaded region in the graph represents where the signal is very close to the background and is excluded from the fit. This data was fit by the original authors to a similar model which accounted for regulation through stabilization only and also describes the data well. Our goal here is to demonstrate the range of this formalism for

**Table 1. Characterization of phase-space for $FC_{\text{max}}$ from the thermodynamic and full kinetic model.**

| Phase space for $FC_{\text{max}}$ | | | | |
|---|---|---|---|---|
| Model | Tuning $P$ through | FC $(R \rightarrow \infty, P \rightarrow 0)$ | FC $(R \rightarrow \infty, P \rightarrow \infty)$ | Line of separation |
| Thermo – dynamic | $k_{\text{off,P}}$ or $k_{\text{on,P}}$ | $\alpha\beta$ | $\alpha$ | $\beta = 1$ |
| Kinetic – model | $k_{\text{off,P}}$ | $\alpha\beta$ | $\frac{\alpha(N_P + r)}{N_P + \alpha r}$ | $\beta = (N_P + r)(N_P + \alpha r)^{-1}$ |
| | $k_{\text{on,P}}$ | $\frac{\alpha\beta(1 + V_2)}{1 + \alpha\beta V_2}$ | $\alpha$ | $\beta = (1 + V_2(1 - \alpha))^{-1}$ |

parameterizing regulation by TFs with disparate function as a general formalism for characterizing and parameterizing TF function in a less context-dependent way than "activator" or "repressor".

**Regulatory behaviors at finite TF concentration.**  To this point, we have discussed the dependence of the maximum fold-change possible from a TF, $FC_{max}$, which will be observed when TF concentration is saturating. However, we also predict interesting phenomena at non-saturating TF concentrations. Most notably, although the maximum fold-change of a TF is a strictly increasing function with $\alpha$ and $\beta$, that is to say higher $\alpha$ and $\beta$ always leads to an increase in the maximum fold-change observable from the regulated gene, at intermediate concentrations this is not necessarily the case. Fig 3A shows the parametric plot of fold-change, FC as a function of $\alpha$ and $\beta$ but with a fixed, sub-saturating concentration of TF. Most notably, the top left corner of this plot which corresponds to low $\alpha$ and large $\beta$ shows the minimum expected fold-change, even lower than expected for TFs whose regulatory role is both destabilizing and decelerating. This counter-intuitive result occurs because at finite TF concentrations, the promoter can express from both the RNAP-only bound state and the co-bound state. In these cases, the TF-RNAP co-bound state becomes extremely stable ($\beta \gg 1$) and transcriptionally unproductive ($\alpha \ll 1$); this specific combination of parameters locks the gene into a low activity state. When the complex is less stable ($\beta < 1$), it will allow for relatively increased expression from the RNAP-only bound state. Fig 3B shows contours of FC vs $\beta$ at fixed $\alpha$. As can be seen, below a certain value of $\alpha$, higher stabilization actually decreases the

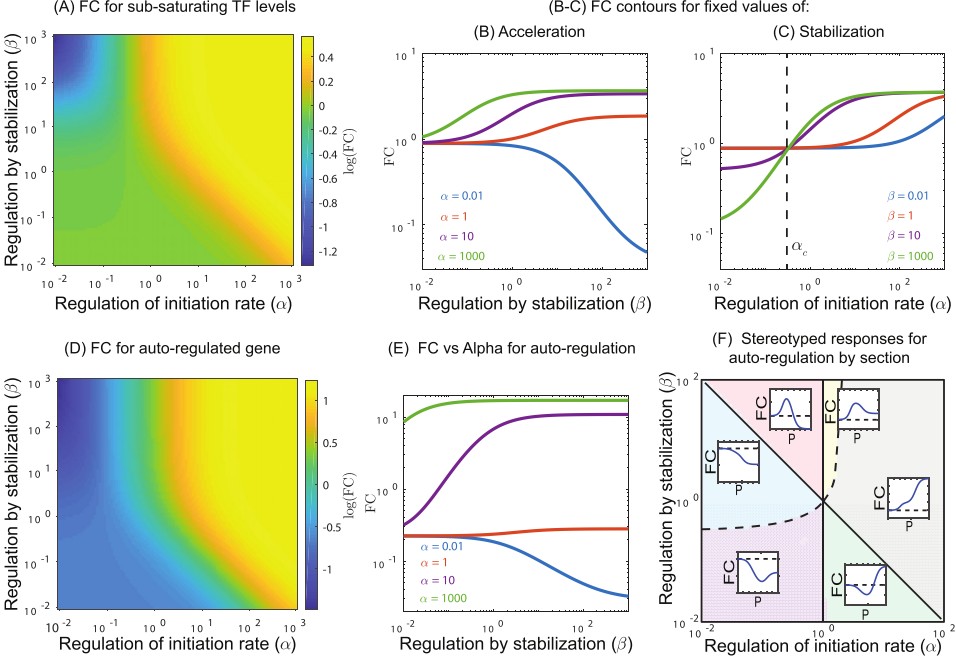

**Fig 3. Regulation at non-saturating levels of TF.** (A) Fold-change in expression as a function of $\alpha$ and $\beta$ at a sub-saturating TF concentration, $R = 0.2$. Here we see a curious feature where the upper left part of the phase space has the lowest fold-change. (B-C) Increasing the stabilization, $\beta$, increases fold-change for values of $\alpha$ above a critical value $\alpha_c$, however, when $\alpha < \alpha_c$, the fold-change decreases with increased stabilization. The critical value, $\alpha_c$, can be exactly solved for and depends on $P$ and $R$. (D-E) This phenomenon persists when examining autoregulatory genes. (F) Although the boundaries between regions of regulation space change for autoregulation, the same qualitative regions are seen, with the primary difference that $FC \to 1$ as $P \to 0$.

fold-change. We can derive a simple relationship that determines if increased stabilization ($\beta$) will increase or decrease fold-change and the point of inflection between these two regimes occurs when $\alpha$ is a specific value $\alpha_c = P/(1 + P + R)$ (derived by setting the derivative of fold-change from the thermodynamic model with respect to $\beta$ to zero. see S1 Fig for comparison between the thermodynamic and full model). When TFs are saturating ($R \to \infty$) the inflection point occurs at $\alpha = 0$ which is the allowable minimum for $\alpha$ and not realizable, and this is why we did not see this behavior in $FC_{max}$ (Fig 3A). However, when TF is at sub-saturating concentrations, this inflection point occurs at realizable levels of deceleration. Importantly, $\alpha_c$ is strictly decelerating, this phenomenon is not possible when the interaction is accelerating $\alpha > 1$. Regardless, for a finite TF concentration ($R$) if $\alpha$ is less than $\alpha_c$ a TF with higher stabilization ($\beta$) produces a lower fold-change than a TF with lower stabilization. On the contrary, if $\alpha$ is greater than $\alpha_c$ then higher stabilization values will lead to a higher fold-change (see Fig 3C).

**Regulatory behaviors of auto-regulation.** We also consider the same simple regulatory architecture, a gene regulated only by one TF, but in the case where the gene itself produces the TF, *i.e.*, an autoregulated gene. In this case, $N_{TF}$ is no longer a parameter that we control but rather it is determined by the characteristics of the gene, the affinity of the TF for the binding site, the strength of the promoter $P$ and the parameters of the regulatory interactions, $\alpha$ and $\beta$. The phase space of FC as a function of $\alpha$ and $\beta$, shown in Fig 3D looks similar to that for finite TF number shown in panel A of the same figure. Although there are some quantitative differences. Interestingly, the same phenomenon occurs in autoregulation where a critical value of acceleration $\alpha_c$ exists, below which increased stabilization leads to lower fold-changes (Fig 3E) Although, the analytic value of $\alpha_c$ is no longer the same and we do not have a simplified expression as in the case of simple regulation, we estimate $\alpha_c$ numerically and compare it with the full model for various promoter strength (see S1 Fig). Finally, we can again divide the parametric space of $\alpha$ and $\beta$ into regions with qualitatively different responses to changing promoter strength $P$ (Fig 3F). Here we show that the same six regions exist that we observed previously for non-autoregulation (Fig 2C). However, the line $\beta = 1$ no longer divides the regions of behavior, the dividing line severely reduces the size of the yellow region (strong activation that gets weaker with increased $P$) and moderately cuts into the purple region (strong repression that gets weaker with increased $P$). In addition, there is now a non-monotonic behavior that is observed in many of these curves unlike in simple regulation (Fig 2D). This is because in simple regulation with saturating TFs, as $P$ approaches 0 the fold-change approaches $\alpha\beta$. However, for autoregulation as $P$ approaches 0 the fold-change approaches 1, because a very weak promoter will not make enough TF to reguate the gene.

## Response time

Next, we explore how the response time of a regulated gene is influenced by acceleration and stabilization. In order to compute response time, we assume that at time $t = 0$ the gene is expressing constitutively and at that time, TFs are instantly switched to an active state. Once actively regulated by TFs, the expression level will change before eventually reaching a new steady state. The response time is then computed as the time for the expression to reach halfway from the prior, unregulated level to the new regulated level (see S1 Text section I for details).

We find that when $R$ goes to infinity, the response times do not depend on the regulatory parameters $\alpha$ and $\beta$; the response time is one-cell cycle (S2B Fig). When $R$ approaches zero, the new steady state also approaches to the constitutive level and hence we cannot determine the response time. However, for intermediate values of $R$, the response time depends on the properties of the regulating TF. As an example, Fig 4A shows three curves of expression as a

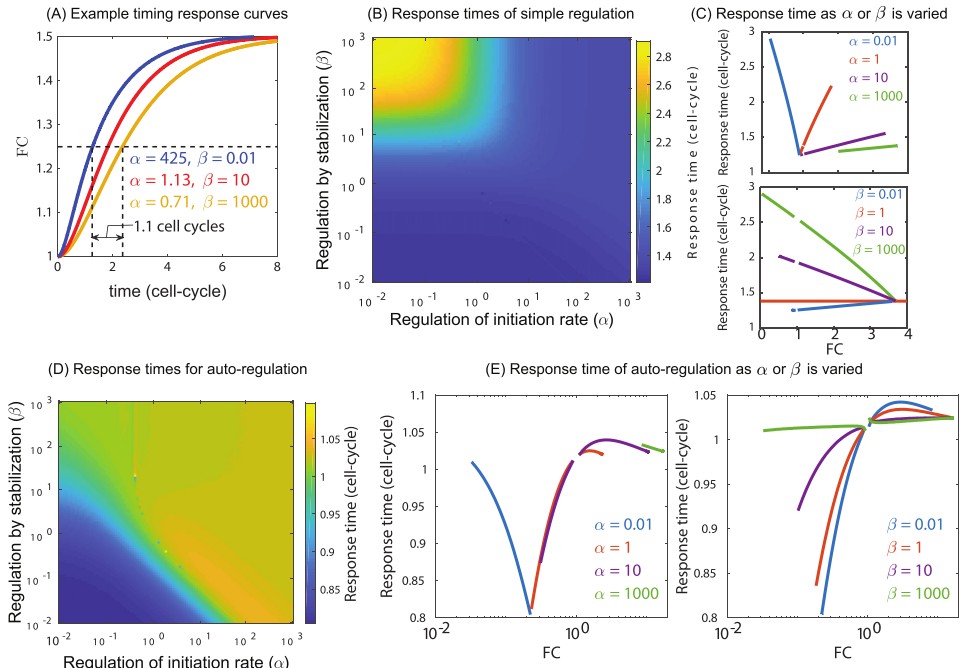

**Fig 4. Response time of gene regulation depends on regulatory parameters.** (A) Example traces of three TFs that have the same mean level of expression (FC) of a regulated gene at steady state. The time it takes to reach 50% of steady state varies substantially depending on the regulatory mechanism used by the TF. (B) Response time as a function of the parameters $\alpha$ and $\beta$. (C) Response time as a function of FC for TFs with a fixed $\beta$ or $\alpha$ as the other parameter is varied. (D) Response times in the case of auto-regulation are significantly less variable (see scale). (E) Response time of auto-regulatory genes as a function of FC for fixed $\alpha$ or $\beta$ while the other parameter is varied. Gaps in the data curves occur because response time becomes meaningless as FC approaches 1.

function of time for different values of $\alpha$ and $\beta$ having the same steady state fold-change 1.5. We find that for the same fold-change faster response time can be achieved by a TF having large $\alpha$(425) and small $\beta$(0.01) (blue curve). For the same fold-change with $\alpha$ = 0.7 and $\beta$ = 1000, the response time becomes slower by over one cell-cycle (yellow curve). A heat map showing the response time in $\alpha - \beta$ space is shown in Fig 4B. We find that for strong stabiliza-tion ($\beta > 1$) and deceleration ($\alpha < 1$) the response times are the longest and reach several cell-cycles. Intuitively this is related to the result in Fig 3A, where fold-change is lowest given these same parameters, caused by long-lived, unproductive TF-RNAP complexes at the promoter which will result in longer wait times on average. A useful way to compare the response time is to examine response times of regulation as a function of fold-change. In Fig 4C, we plot response time as a function of fold-change tuned by changing the stabilization, $\beta$ at fixed $\alpha$ (top panel) or by changing acceleration $\alpha$ at fixed stabilization $\beta$. In the bottom panel of Fig 4C, we see that for the same mean fold-change, TFs with lower $\beta$ values (less stabilizing) have lower response times. In particular, depending on the value of $\beta$ the response time decreases or increases monotonically with the fold-change. When the TF is a stabilizer ($\beta > 1$), increasing the fold-change decreases the response time. On the other hand, when the TF destabilizes ($\beta <$ 1) increasing the fold-change increases the response time. Importantly, when $\beta = 1$ the response time is independent of the fold-change and is set by the constitutive gene (brown curve). In Fig 3C we have shown that the fold-change is always monotonically increasing with $\alpha$. This implies that when the TF is a stabilizer increasing $\alpha$ will lead to higher fold-change

with faster response times. On the contrary, when the TF is a destabilizer increasing $\alpha$ will lead to higher fold-change with slower response times (Fig 4C).

In the top panel of Fig 4C we examine the response time when $\alpha$ is fixed and $\beta$ is tuned to explore the response time as a function of fold-change. From Fig 3B, we know that when $\alpha$ is below a critical value, increasing $\beta$ will cause a decrease in fold-change. However, when $\alpha$ is above this threshold, increasing $\beta$ will cause fold-change to increase. This is seen in the top panel of Fig 4C, however, regardless of the specific value of $\alpha$, the response time increases with $\beta$. This also implies that when TF is a repressor increasing fold-change reduces the response time (blue curve) and when the TF is an activator increasing fold-change increases the response time (green, purple, and brown curves).

We also examine how the response time depends on the regulatory parameters for an auto-regulated gene. For an auto-regulated gene the number of TFs is a function of the regulatory parameters and hence the number of TFs are not fixed as we probe different $\alpha$ and $\beta$. As before, we assume that at time $t = 0$ the gene is expressing constitutively and at that time, TFs are instantly switched to an active state, i.e., the protein product of the gene binds to its own DNA and start regulating the gene. A heat map showing the response time in $\alpha - \beta$ space is shown in Fig 4D. We find that for weak stabilization and weak acceleration the response times are the fastest and are lower than a cell-cycles. In fact, for the parameter regime explored we find that the response times hardly exceeds one cell cycle irrespective of the TF identity, an activator or a repressor. We also find an intermediate maximum in the response time as a function of both stabilization and acceleration which can be viewed in response time versus fold-change plots (Fig 4E).

## Noise in expression

We compute noise as the coefficient of variation (CV) of the expression defined as the ratio of the standard deviation to the mean. In general, we will consider the fold-change in CV, which compares the change in noise from regulation to that of the unregulated ($R = 0$) case. For finite TF concentration the expression for CV is intractable and we solve it numerically (see S1 Text section IV). However when $R \to \infty$ the expression for noise ($CV_{max}$) reduces to a simple form and is given by

$$\text{Fold} - \text{change in CV} = \frac{CV_{max}}{CV_0} = FC_{max}^{-1/2} \left( \frac{Fano_{max}}{Fano_0} \right)^{1/2}. \tag{4}$$

where, the fano-factor, defined as the ratio of variance to the mean, is

$$Fano_{max} = 1 - \frac{\alpha\beta V_2 P}{(1 + \alpha\beta V_2 + \beta P + \beta V_4)(1 + \alpha\beta V_2 + \beta P)}. \tag{5}$$

In the above expressions, setting $\alpha = \beta = 1$ gives the noise of a constitutive gene ($CV_0$ and $Fano_0$). In Fig 5, we examine how the noise is modulated as a function of TF concentration for various regulatory parameters and promoter strengths. For a pair of ($\alpha, \beta$) and promoter strength ($P$) we vary the TF number in a range from ($0, \infty$) and examine the dependence of noise on the TF concentrations. When the number of TF is infinite the fold-change in CV approximately follows inverse square root of fold-change in expression shown as black line in Fig 5B. In Fig 5A, the solid lines show the trajectory of a typical repressing TF interactions as the number of TFs is titrated, while the dashed lines show activating interactions as the number of TFs is titrated for various promoter strengths and regulatory parameter $\alpha, \beta$. We find three typical behaviors of the relationship between FC in CV and TF copy number. The first, colored blue, are cases where FC in CV is monotonically increasing (for repressing

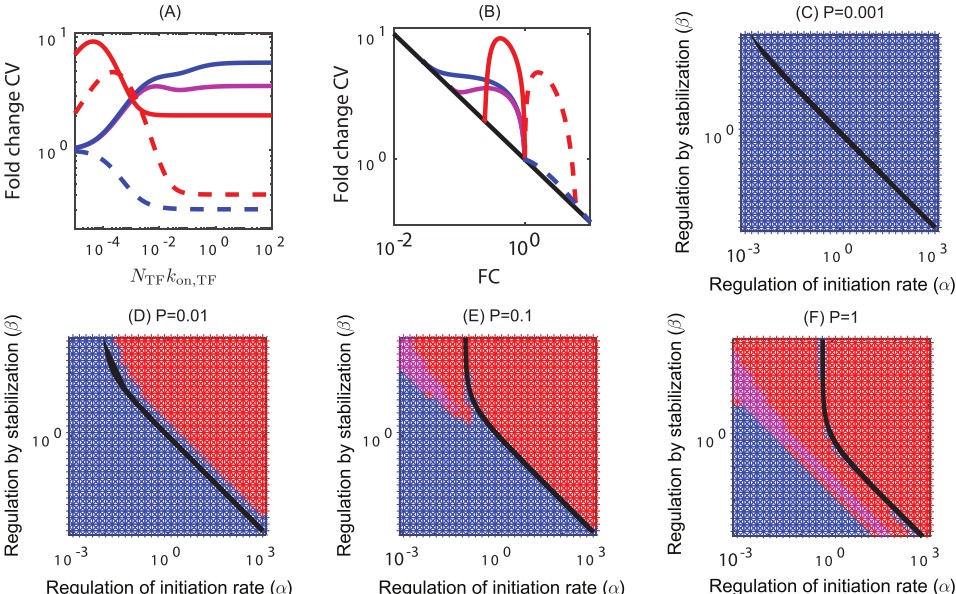

**Fig 5. Phase-space showing monotonic and non-monotonic noise in gene expression as a function of TF concentration for various regulatory parameters.** (A) The fold-change in CV as a function of TF concentration, $k_{\text{on,TF}}$ can be monotonic (blue) (*i.e.* noise is highest for saturated TF concentrations), non-monotonic with a maximum at finite TF concentration (red), or non-monotonic with an internal maximum and minimum (magenta). The shape of this curve depends on both the promoter strength (*P*) and the regulatory parameters ($\alpha$, $\beta$) of the TF. (C-F) The phase-space of noise for promoters of various strength. The color of the space in these plots specifies the qualitative shape of the fold-change in CV vs $k_{\text{on,TF}}$ plot. The black line separates repressive TFs (left of the line) and activating TFs (right of the line). (C) For weak promoters, the noise is always monotonic. (D-F) For stronger promoters we see that activating TFs are always non-monotonic when the TF is activating (points to the right of the black line). However, in the repression regime, the noise can be monotonic (blue), or non-monotonic with a maximum (red) or both maximum and a minimum (magenta).

interactions) or decreasing (for activating interactions). The second type, colored red, show an intermediate maximum in FC in CV at a finite concentration of TFs. The last color, magenta, marks relationship where the FC in CV has both a maximum and a minimum at intermediate concentration. In some cases, the intermediate peak is a global maximum while in other cases the value at TF $\rightarrow\infty$ is greater than the intermediate peak. The occurrence of these relationships is shown in Fig 5C–5F for 4 different promoter strengths. We find that for very weak promoter the noise is always monotonic irrespective of whether it is an activator or a repressor (see Fig 5C). For such promoters the noise is always higher than the constitutive gene if the TF is a repressor. On the contrary, the noise is always lower than the constitutive gene if the TF is an activator, i.e., the noise decreases monotonically with TF concentration and saturates at $CV_{\text{max}}$. As we increase the promoter strength, the activators show a non-monotonic behavior (shown as red points in Fig 5D–5F) with an intermediate peak in the upper right corner of the phase-space which is greater than the noise of a constitutive gene. The regulatory parameters in the vicinity of $\alpha\beta = 1$ still show a monotonic noise. For a strong promoter, we find that the activators always show a non-monotonic behavior with an intermediate peak. However, the regulatory parameters corresponding to repressors show 3 different regimes (Fig 5F); low $\alpha$ and $\beta$ regime with monotonic noise (blue), a non-monotonic regime with a maximum (red), and a regime sandwiched in between where both a maximum and a minimum in noise is observed (magenta).

## Conclusion and discussion

Here we have considered a simple model of TF function that accounts for regulatory interactions on two steps in gene expression process: occupancy of RNAP at the promoter and initiation of the transcription process. The realization of this model is that the characteristics of even the simplest regulatory systems are sensitive to the interplay of the regulatory interactions and the relative strength of the promoter. Primarily this interplay arises because the relative "value" of stabilizing interactions to a promoter depends on how well the promoter is capable of recruiting polymerase in the absence of TF; stronger promoters are altered less by stabilization than weak promoters. Although it is not in our model, acceleration should have a similar limitation set by the speed of transcriptional elongation. Our model assumes nothing about the relationship between these characteristics, only that they exist. In practice it may be that these modes are correlated in some way, for instance, perhaps the fundamental interactions that produce stabilizing interactions also tend to be decelerating. However, in a past study we found stabilizing TFs with both accelerating and decelerating interactions; the methodology in that study was unable to resolve destabilizing interactions [39]. The use of *in vitro* approaches to measure fold-changes of the kinetic steps of transcription for individual TFs would provide means for independently determining regulatory parameters, and would allow for direct tests of this model with *in vivo* regulation data. Here we have used ranges for $\alpha$ and $\beta$ which result in net fold-changes that are consistent with the range of regulatory effects we see experimentally using the single-TF titration library [19]. Although this model is written with bacterial gene regulation in mind, similar approaches have been useful in describing TF function in eukaryotic systems [18, 22]. In particular, the idea of kinetic control from "specialized" TFs that work on distinct kinetic steps [23, 38, 52] has demonstrated how TFs can function sequentially to promote gene expression using a kinetic framework [18].

Among the most striking consequences of this model is that the regulatory function of a TF depends strongly on the promoter it is regulating. It is common to classify TFs as "activators" or "repressors" and presume that this is a meaningful classification, however, the implications of this model is that the net regulatory action of a TF is a complex interplay of these two mechanisms and, as such, is not a good classifier of TFs. Even the broad, intuitive characterizations such as "more stabilizing TFs should produce larger fold-changes" fails in certain situations (see Fig 3A). Recently, we have seen that regulation by CpxR appears to be both stabilizing ($\beta > 1$) and decelerating ($\alpha < 1$) for many binding locations (blue and red region of Fig 2D). TFs with such "incoherent" regulatory interactions are particularly prone to be difficult to predict even their qualitative regulatory features on a promoter based on measurements of a different promoter.

Regulatory modes impact more than just the steady-state levels of expression. We show that the response time of a gene to regulation depends on the regulatory modes of the TF. For simple regulation, the heat map of response times almost resembles an "AND" logic gate where a combination of stabilization and deceleration will cause the response of the gene to be slow with the possibility of doubling the response time compared to faster combinations. We do not see this same phenomenon for auto-regulation where the space is considerably more confined, however in all cases auto-repressive circuits respond faster than auto-activation. As a general phenomenon, it is known from simple models that auto-repression speeds up responses while auto-activation slows response times [53, 54], we show here that based on the regulatory parameters of the TF, the response times can vary substantially, although the general phenomenon that auto-repression speeds up responses while auto-activation slows response times is broadly true in this case [54, 55].

We further investigated the impact of the regulatory modes in the noise defined as the coefficient of variation (CV) of gene expression. We find three characteristic feature of noise as a function of fold-change. For weak promoter the noise is always monotonically increasing with the fold-change irrespective of whether the TF is an activator or a repressor. On the other hand for strong promoters, the noise can be monotonically increasing or non-monotonic with a maximum at intermediate fold-change or both a maximum and a minimum depending on the regulatory parameters.

Overall, the balance of two regulatory interactions significantly alters every property of a TFs function. Although there are infinitely many combinations of $\alpha$ and $\beta$ to achieve a given average fold-change, specific values of these parameters will alter the response dynamics of the gene, the noise and even how that TF will regulate other promoters. In this case, the broad labels of "activator" or "repressor" will not capture anything general about a TFs interaction. However, our knowledge of the mechanisms of action of TFs is relatively limited, an important next step is to understand what part of this phase space is actually seen in real TFs.

## Supporting information

**S1 Text. Model and simulation methodology.** (I). Model and simulation methodology. (II) Derivation of Fold-change from Thermodynamic model. (III) Fitting model to pre-existing data. (IV) Noise in gene expression. (V) Autoregulating gene.
(PDF)

**S1 Fig. Critical acceleration, $\alpha_c$, for different models of gene regulation.** (A) $\alpha_c$ from the thermodynamic model (black lines) and the full model (red lines) as a function of TF concentration for weak ($P = 0.01$) and strong promoter ($P = 1$). In order to find $\alpha_c$ for the full model, we compute fold-change as a function of $\beta$ for a range of $\alpha$ values with small increments. We then find the value of alpha where the fold-change vs $\beta$ curve switches from monotonically decreasing to monotonically increasing. (B) $\alpha_c$ for autoregulated gene (blue line) and the genes regulated by fixed TF concentration (red lines) as a function of promoter strength.
(PDF)

**S2 Fig. Response time when TF concentration is tuned.** Response time as a function of fold-change (A) or TF concentration (B) as the TF concentration is tuned. As the TF concentration approaches to infinity the response time is saturated to one cell-cycle irrespective of the regulatory parameter of the TF.
(PDF)

**S3 Fig. Behavior of auto-regulated genes when TFs act as a dimer.** Fold-change (A,C) and response time (B,D) of autoregulating gene for slow dimerization ($k_{dim} = 0.1s^{-1}TF^{-1}$, $k_{mon} = 1s^{-1}$, Panel A,B) and fast dimerization ($k_{dim} = 1s^{-1}TF^{-1}$, $k_{mon} = 0.1s^{-1}$, Panel C,D). (E,F) The fold-change of auto-regulating gene versus titration of promoter strength. The qualitative feature of TF switching from activator to repressor when $\alpha < 1$ and $\alpha\beta > 1$ (yellow curve in E) as well as switching from repressor to activator when $\alpha > 1$ and $\alpha\beta < 1$ is preserved (magenta curve in F).
(PDF)

## Author Contributions

**Conceptualization:** Md Zulfikar Ali, Vinuselvi Parisutham, Robert C. Brewster.

**Data curation:** Md Zulfikar Ali.

**Formal analysis:** Md Zulfikar Ali, Sunil Guharajan, Vinuselvi Parisutham.

**Funding acquisition:** Robert C. Brewster.

**Investigation:** Md Zulfikar Ali, Sunil Guharajan, Vinuselvi Parisutham, Robert C. Brewster.

**Methodology:** Md Zulfikar Ali, Robert C. Brewster.

**Project administration:** Robert C. Brewster.

**Software:** Md Zulfikar Ali, Sunil Guharajan.

**Supervision:** Robert C. Brewster.

**Validation:** Md Zulfikar Ali.

**Writing – original draft:** Md Zulfikar Ali, Robert C. Brewster.

**Writing – review & editing:** Md Zulfikar Ali, Sunil Guharajan, Vinuselvi Parisutham, Robert C. Brewster.

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
