## [Decision Letter · Decision Letter 0]

28 Mar 2024

Dear Professor Brewster,

Thank you very much for submitting your manuscript "Regulatory properties of transcription factors with diverse mechanistic function" for consideration at PLOS Computational Biology.

As with all papers reviewed by the journal, your manuscript was reviewed by members of the editorial board and by several independent reviewers. In light of the reviews (below this email), we would like to invite the resubmission of a significantly-revised version that takes into account the reviewers' comments.

All 3 reviewers are basically positive about the paper, but make extensive comments how it can be improved. Please consider their suggestions.

We cannot make any decision about publication until we have seen the revised manuscript and your response to the reviewers' comments. Your revised manuscript is also likely to be sent to reviewers for further evaluation.

Sincerely,

Stefan Klumpp

Academic Editor

PLOS Computational Biology

Pedro Mendes

Section Editor

PLOS Computational Biology

All 3 reviewers are basically positive about the paper, but make extensive comments how it can be improved. Please consider their suggestion.

Reviewer's Responses to Questions

**Comments to the Authors:**

Reviewer #1: Major concerns:

In the manuscript, the authors describe their model as using the simplest regulatory architecture, meaning that the promoter is regulated only by a single TF and that the TF has only one binding site on the promoter. It is said, that modeling approaches are a focus in the field of regulatory biology. Are there others/similar models available in the literature and what are the pros/cons and limitations compared to the model presented?

Is it possible to apply the model to literature data as it was done before in their previous work for Escherichia coli to verify the results?

When introducing the fold-change in Equation 1, you reference to [39]. In [39] Fig. 1A you use a different definition for FCmax. Could you explain where the difference in the model comes from?

Minor concerns:

In Fig. 1B, it would be nice to indicate the role of β by highlighting the altered rate.

"The TF and the polymerase unbind/dissociate from the bound states at the rate that is independent of the TF/polymerase concentration and is only dependent on the interaction between

the TF and the polymerase, and binding site identity. We denote the unbinding rates of the TF and polymerase as kTF and kP, respectively."

In Fig. 1B: If the unbinding rate is dependent on the interaction between the TF and the polymerase, why are kTF and kP the same for the transitions from either TF bound or polymerase bound to the empty state and the TF and polymerase co-bound state to the other states? Why is the unbinding rate not different coming from the co-bound state?

On page 3 you reference (Fig. 2B, right panel). Is this supposed to be Fig. 2C?

On page 4: "Here, V2 = r/kP is the ratio of the transcription rate from polymerase bound state and the polymerase unbinding rate without TF (kP,off)."

In the equation for V2, is it supposed to be kP,off or kP?

Page 5: "A heat map showing the response time in α − β space is shown in Fig. 4B. We find that for

strong stabilization and deceleration (α < 1) the response times are the longest and reach several cell-cycles."

Just for clarity it would be nice to add (β > 1) after strong stabilization.

"Code and results are available on Github." - I could not find a link anywhere.

Reviewer #2: The study builds on a model proposed and studied by the authors in ref. 39 for the simplest form of gene regulatory architecture – one promoter is regulated by a single transcription factor that has only one binding site on the promoter of a target gene. The model uses two parameters – alpha and beta, with alpha quantifying the changes to the rate of open complex formation and beta quantifying the changes in the RNA polymerase occupancy at the promoter; both parameters take values from 0 to infinity, with 1 the point that marks the difference between accelerating / decelerating (alpha) and stabilizing / destabilizing (beta). The main insights obtained from the model is that the regulatory role of a TF depends strongly on the promoter it regulates. These insights are further supported by looking into response times, auto-regulation, and noise.

The manuscript is well-written and easy to follow, with instructive figures that help with illustration of the main points. The reviewer has few minor points that can help with bridging the gap between the simplifying assumptions made in the model and the reality of gene regulation (even in prokaryotes).

Comments

1. The model, as presented, holds for prokaryotes. It would be great if the authors include few points of discussion about what modifications of the model, at minimum, would be needed to render it applicable to study gene regulation in eukaryotic systems.

2. Could the authors comment on the effects of having a promoter regulated by more than one transcription factor? To what extent could any of the conclusions from the simple model hold? This is relevant for establishing the relation between model and actual regulation in prokaryotic systems.

3. Could authors comment how the numerical estimation of the parameter alpha_c. This was not explained in the methods or the main part of the text.

4. Fig. 1, spell out Txn rate on panel C.

Reviewer #3: In this manuscript, Ali et al. predict general properties of transcription factors based on their interactions with their binding site and RNA polymerase, as well as the strength of the promoter they regulate. The authors analyze a clever model designed in a previous publication from their lab, where the effect of a transcription factor is divided into two parts: the recruitment of RNA polymerase and transcription initiation. The analysis of this model results in unexpected results, such as parameter regimes where transcription factors can behave as either repressors or activators depending on the strength of the promoter. The authors also calculate general properties of these regulators, such as response times and noise. Overall, this manuscript presents an interesting analysis that advances our understanding of the mechanics of transcriptional regulation. The manuscript is clearly written (although a thorough check is advisable), and this research is of interest to the wider community. However, since the model is not new, this reviewer believes there are important pieces missing in the analysis, and some key findings of the manuscript should be clarified. Ultimately, I am favorable for publication provided the authors can address the following points:

Major points:

- This manuscript needs a discussion on realistic ranges of parameters for transcription factors found in nature. The example values are often orders of magnitude apart, so it might be helpful to discuss whether these parameters could plausibly have such a wide range. Ideally, the authors should find examples where stabilization and acceleration can be calculated from existing data. Equilibrium constants of the TF-RNAP-DNA complex or transcription initiation rates might not be readily available, but at the minimum the authors should comment on possible means to measure stabilization and acceleration, as well as comment on what are reasonable ranges expected to be found in nature. Recent studies on the properties of the parameter space of transcriptional regulation did a similar analysis (PMID: 36168291).

- Although the model of gene regulation by a single TF yields interesting results, the manuscript does not address the effects of cooperativity, which is the norm in transcriptional regulation. The addition of such nonlinearities to the model is particularly relevant to the case of self-regulation, and could potentially reveal new properties. Implementation of dimerized binding of transcription factors within this framework is relatively straightforward, and, in my opinion, would significantly expand the scope of the results. But at least a discussion would be very useful.

- Some of the key findings of this manuscript are not obvious, and would greatly benefit from more intuitive explanations. The authors provide such explanation when showing why incoherent promoters can be activators or repressors depending on promoter strength. However, it is still not obvious to me why higher stabilization decreases fold-change at finite TF concentrations with low acceleration. Those conditions also seem to result in the longest response times. Similarly, there are other results throughout the paper that could be better interpreted.

Minor points:

- The derivation of the thermodynamic model should be better explained and discussed, perhaps a section in the supporting material would help. For instance, it is not immediately obvious to me why the effective concentration of the TF (chi) depends on beta. Presumably the complex formed with RNAP stabilizes the binding of the TF itself, as can be inferred from the panel in Fig 1C (it could be the case that TF binding does not depend on RNAP presence). However, this is not discussed in the text, and there is no reference to beta in the diagram in Fig 1B.

- In the Introduction, the authors mention “Our model is primarily suited for prokaryotic gene regulation, however it can be generalized for eukaryotic systems as well.” What aspects of eukaryotic gene regulation would need to be added to their modeling framework?

- In Fig 1, the kinetic model precedes the thermodynamic model. It would be useful to present the models in the order they are cited.

- In the Model section, R should be defined when it’s introduced.

- In the second paragraph of the section on the phase space approach: “As beta becomes larger, FCmax simply becomes alpha*(P+1)/P”.

- The way the main text references Figure 2 is inconsistent with how it’s laid out. The text refers to the left and right panels, but the figure has them labeled as B and C.

- In page 4, the authors state that “all results that follow will be derived from the full kinetic model”, and then proceed to derive results from the thermodynamic model. More clarity on the model used would help.

- In page 4, the authors discuss that the thermodynamic model applies in the limit where the transcription rate is much slower than RNAP unbinding. Is that generally true? I would expect it to be, but a discussion on realistic parameters would help.

- In page 5: “where a critical value of acceleration alpha_c exists”.

- In Fig 3D-F and Fig 4DE, it would be helpful to show what is the resulting equilibrium R when auto-regulation is introduced. Especially when the phase diagram differs in the two scenarios.

- In Fig 3F, it would be helpful to delineate the parameter space according to the different regions (perhaps a dashed line for beta=1?).

- In Fig 3F, FC is no longer monotonic in P. The authors should comment on what is the cause of that.

- Fig 4E seems to suggest that, in certain parameter ranges, response time is less sensitive to parameter alpha than in the case of a TF that doesn’t regulate itself. For example, the curves of response time vs FC for alpha>1 are practically on top of each other. A discussion on how this is achieved by self-regulation would be helpful. Also, those curves should have their gaps fixed.

- Technically, dilution of cell components is due to cell growth, not division.

- In the section about noise, why vary k_TFon as a proxy for TF number?

- In “Noise in expression,” consider including a brief definition/explanation of the Fano factor, as readers may be less familiar with it.

- Overall, I am confused about the magenta regions on Fig 5. It looks very similar to the blue regions, and I don’t quite understand what is the basis for the distinction. An explanation on why the FC in CV shows both local maxima and minima would be helpful.

- In the conclusion section, a reference to an “AND logic gate” to refer to response times sounds confusing to me.

**Have the authors made all data and (if applicable) computational code underlying the findings in their manuscript fully available?**

Reviewer #1: **No: **The Github link is missing.

Reviewer #2: **No: **No code is provided with the manuscript. Github link could be included.

Reviewer #3: **No: **The authors state that the code is available on GitHub, but I could not find the link.

PLOS authors have the option to publish the peer review history of their article (what does this mean?). If published, this will include your full peer review and any attached files.

Reviewer #1: No

Reviewer #2: No

Reviewer #3: No

Figure Files:

While revising your submission, please upload your figure files to the Prefl

---

## [Decision Letter · Decision Letter 1]

24 May 2024

Dear Professor Brewster,

We are pleased to inform you that your manuscript 'Regulatory properties of transcription factors with diverse mechanistic function' has been provisionally accepted for publication in PLOS Computational Biology.

Best regards,

Stefan Klumpp

Academic Editor

PLOS Computational Biology

Pedro Mendes

Section Editor

PLOS Computational Biology

Reviewer's Responses to Questions

**Comments to the Authors:**

Reviewer #1: The authors have addressed all of my concerns sufficiently.

**Have the authors made all data and (if applicable) computational code underlying the findings in their manuscript fully available?**

Reviewer #1: Yes

PLOS authors have the option to publish the peer review history of their article (what does this mean?). If published, this will include your full peer review and any attached files.

Reviewer #1: No

---

## [Editor Report · Acceptance letter]

4 Jun 2024

PCOMPBIOL-D-24-00107R1 

Regulatory properties of transcription factors with diverse mechanistic function

Dear Dr Brewster,

I am pleased to inform you that your manuscript has been formally accepted for publication in PLOS Computational Biology. Your manuscript is now with our production department and you will be notified of the publication date in due course.

With kind regards,

Zsofia Freund
